# Fungal Diversity Profiles in Pit Mud Samples from Chinese Strong-Flavour Liquor Pit

**DOI:** 10.3390/foods11223544

**Published:** 2022-11-08

**Authors:** Shunchang Pu, Shoubao Yan

**Affiliations:** 1Department of Biology and Food Engineering, Bozhou University, Bozhou 236800, China; 2College of Biological Engineering, Huainan Normal University, Huainan 230038, China

**Keywords:** fungal community, pit mud, volatile flavor compounds, fermentation

## Abstract

Pit mud, a specific fermented soil, is an essential material for the fermentation of Chinese strong-flavour liquor. However, few studies to date have sought to characterize the spatial profiles of pit mud fungal communities in fermentation cellars from Chinese strong-flavour liquor distilleries. In this analysis, differences in fungal community structures and physicochemical properties in pit mud samples from different spatial positions within fermentation cellars were analyzed, revealing unique characteristic multidimensional pit mud fungal community profiles. *Penicillium roqueforti*, *Pichia kudriavzevii*, *Aotearoamyces nothofagi*, *Penicillium robsamsonii*, *Alternaria arborescens*, *Trichosporon insectorum*, *Seltsamia ulmi*, *Trichosporon coremiiforme*, *Malassezia restricta* were dominant in the pit mud samples form the upper cellar wall, whereas *Metarhizium frigidum*, *Calonectria pseudoreteaudii*, *Penicillium clavigerum*, *Fusarium equiseti*, *Simplicillium chinense*, *Aspergillus intermedius*, *Trichosporon coremiiforme*, *Fusarium circinatum*, *Alternaria radicina*, *Aspergillus heterocaryoticus* were predominant in the middle cellar wall. *Alternaria radicina*, *Cladosporium chasmanthicola*, *Alternaria helianthiinficiens*, *Penicillium argentinense*, *Antarctomyces psychrotrophicus*, and *Trichosporon inkin* are majorly present in the down cellar wall layer. *Bipolaris axonopicola*, *Ramgea ozimecii*, *Penicillium argentinense*, *Calonectria queenslandica*, *Metarhizium robertsii*, and *Penicillium roqueforti* were identified as the dominant fungi in pit mud samples from the cellar bottom. Additionally, *Alternaria destruens* and *Alternaria doliconidium* are present at notably high levels in all layers of pit mud samples. Moisture, pH, PO_4_^3−^, acetic acid, humus, K^+^, Mg^2+^, Ca^2+^, butyric acid, and caproic acid levels in these different pit mud positions exhibited a rising incremental pattern from the upper wall layer to the bottom layer, whereas lactic acid levels were significantly lower in the bottom pit mud layer relative to these other layers. Moisture, pH, and NH_4_^+^-N were identified as the three most significant factors associated with fungal community composition through a redundancy analysis. Overall, these findings may offer a theoretical foundation for future efforts to improve or standardize artificial pit mud.

## 1. Introduction

Chinese strong-flavour liquor is a traditional fermented beverage that accounts for roughly 70% of total liquor consumption in China [1]. Owing to its unique flavour and brewing approach, strong-flavour liquor holds a special status in Chinese culture and history. This liquor is distilled in large rectangular pit cellars (3600 × 2300 mm at the top; 2800 × 1540 mm at the bottom; 2400 mm deep) that serve as fermentation vessels (Figure 1). The walls of these pits are covered with a specific type of fermented clay known as pit mud that contains large quantities of functional microorganisms including *Clostridium* spp., *Bacillus* spp., and *Methanobacterium* spp., all of which serve as key mediators of the fermentation process and sources of the aromatic compounds characteristic of Chinese strong-flavour liquor [2]. Indeed, the microbes within pit mud are generally accepted to play an essential role in the process of Chinese strong-flavour liquor fermentation [3]. Given their importance, many studies have analyzed these microbial communities in an effort to better understand the mechanisms whereby these organisms contribute to the liquor production process [4]. 

Studies of pit mud conducted to date have primarily focused on prokaryotic flora [2]. For example, Xu et al. investigated the prokaryotic community succession in the vertical dimension of Wenwang Chinese strong-flavor *Baijiu* pit muds, and the results show that the depth of 4 cm was the dividing line between shallow and deep pit mud [5]. Hydrogenispor occupied the dominant position in deep pit mud. In the new pit mud, *Lactobacillus* played a dominant role in both shallow and deep pit mud. The relative abundance of *Bacillus* and *Clostridium sensu stricto* 12 increased with the increase of pit mud depth. Wu et al. investigated the relationship between physicochemical properties, microbiome, and volatiles of pit mud at different ages, and the result showed that moisture, available phosphorus and potassium, and the key flavour volatiles (caproic acid, ethyl caproate, and ethyl butyrate) in mature pt mud were higher than those in growing pt mud, whereas lactic acid and hexanol showed the opposite trend. *Proteiniphilum* and *Petrimonas* (*Bacteroidia*) were most abundant in growing pit mud, whereas *Caproiciproducens* and *Clostridium* (*Clostridia*) were most abundant in mature pit mud [6]. The relative abundance of *Clostridia* (*Caproiciproducens* and *Clostridium*) was positively correlated with the content of caproic acid and ethyl caproate, and negatively correlated with the content of lactic acid. Liu et al. used a DGGE approach to explore *Clostridium* cluster I community diversity in samples of pit mud from cellars of different ages (1, 50, 100, and 400 years), revealing *C. ragsdalei*, *C. ljungdahlii*, *C. autoethanogenum*, and *C. kluyveri* to be the dominant species therein [3]. Liang et al. also employed a combination of PCR-DGGE and qPCR approaches to detect higher levels of *Clostridium* IV species in aged pit mud relative to aging pit mud, which they speculated may be associated with the fact that aged pit mud has a strong aroma whereas aging pit mud does not [7]. Ding et al. also conducted a nested PCR-DGGE-based study of eubacterial community structures in Chinese strong-flavour liquor pit mud and found that community diversity was greater in the bottom of the cellar relative to in the cellar walls [8]. 

Fungi are the essential microbes active in Chinese liquor production, responsible for starch degradation, alcoholic fermentation, and the production of aromatic compounds [9]. However, despite their acknowledged importance, the extent of fungal diversity in pit mud is largely unknown. Previously, morphological identification detected *Mucor*, *Aspergillus*, *Penicillium*, *Monascus*, *Absidia*, *Monilia*, *Sartorya*, and *Ovularia* species in pit mud [10]. Despite the identification of many fungal genera, the fungal compositions reported previously were quite different. These differences may be attributable to many factors, such as differences in different spatial positions of the cellar. Up to now, few studies to date have sought to characterize the spatial profiles of pit mud microbial communities in fermentation cellars from Chinese strong-flavour liquor distilleries. This study was thus designed to evaluate these fungal communities in pit mud samples via a multidimensional DGGE approach and by assessing associated sample physicochemical properties. In so doing, we aim to improve pit mud quality and consistency, and to facilitate the generation of artificial pit mud. by exploring pit mud microbial and physicochemical properties. This study is the first to our knowledge to have explored these multidimensional fungal community distributions of fungal communities and physicochemical properties in different spatial positions of pit mud by using PCR-DGGE methods, which may help to provide a better understanding of this mixed microbial community and its role in Chinese strong-flavour liquor production. 

## 2. Materials and Methods

### 2.1. Pit mud Sample Collection

Samples of pit mud were obtained from ~20-year-old pits from a well-known liquor manufacturer (Anhui Yingjia Distillery Group Co., Ltd., Lu-An, China) located in Luan city, Anhui province, China. Sampling sites are detailed in Figure 1. Sampling was conducted as per a stratified random approach [11]. 

Each sample plot was divided into four different positions (upper layer of the cellar wall, middle layer of the cellar wall, down layer of the cellar wall and the bottom of the cellar), and the pit mud wall samples were collected from the centre of each wall, with approximately 100 g of mud being collected per position and mixed to yield a composite sample. Samples of pit mud from the cellar bottom were collected from the centre of each pit. All samples were collected at a depth of ~5 cm. All samples were well mixed, then transferred to sterile polyethylene bags and stored at −20 °C prior to analysis. 

### 2.2. Physiochemical Property Analyses

Pit mud moisture levels were established by drying samples for 3 h at 115 °C. Pit mud pH values were established with a Mettler Toledo pH meter after diluting sample 1:4 (*w*/*v*) with dH_2_O for 3 h. Pit mud ammonium (NH_4_^+^-N) levels were established via extraction in 10% (*w*/*v*) NaCl at a 1:10 (*w*/*v*) ratio, after which concentrations were measured using a UV spectrophotometer. Acetic acid, butyric acid, and caproic acid were extracted using 15% methanol and quantified via gas chromatography (Agilent 7890, Santa Clara, CA, USA) as described previously [12]. Lactic acid (LA) levels were quantified via ultra-high-performance liquid chromatography (UPLC, Acquity I-class, Waters, Milford, MA, USA) as previously reported [1]. Levels of K^+^, PO_4_^3−^, soluble Mg^2+^, and soluble Ca^2+^ in air-dried pit mud were measured via extracting samples with ddH_2_O at a 1:10 (*w*/*v*) ratio, after which concentrations were measured as reported previously by Zhang et al. using an ion chromatograph (ICS5000^+^, Thermo Fisher, Waltham, MA, USA) equipped with a conductivity detector (ICS-5000^+^-DC) and a CS12 column (Ion Pac, Thermo Fisher, 4 mm × 250 mm) [4]. The utilized injection volume was 25 μL, with methane sulfonic acid (20 mM) as a carrier fluid at a 1 mL/min flow rate, with a column temperature of 30 °C. Humus levels were determined as detailed previously by Shen [13]. 

### 2.3. Fungal Community Assessment

#### 2.3.1. DNA Extraction

A Fast DNA SPIN Kit for Soil (MP Biomedicals, Solon, OH, USA) was used based on provided directions to extract DNA from pit mud samples. Samples were analyzed in triplicate. Briefly, a Mini-Beadbeater (Biospec Products, Bartlesville, OK, USA) was used to homogenize samples for 1 min at 5000 rpm, after which DNA was eluted in TE buffer (10 mM Tris-HCl, 1.0 mM EDTA), diluted 20-fold with this same buffer, and analyzed. 

#### 2.3.2. PCR Amplification

The intergenic transcribed spacer region (ITS) gene was first amplified with the universal ITS1f (5′-CTTGGTCATTTAGAGGAAGTAA-3′) and ITS4 (5′-TCCTCCGCTTATTGATATGC-3′) primers as detailed by Cobo-Díaz et al. [11], after which the nested PCR NS2 (5′-GCTGCGTTCTTCATCGATGC-3′) and the GC-clamp (5′-CGCCCGCCGCGCGCGGCGGGCGGGGCGGGGGCACGGGGGGCTTGGTCATTTAGAGGAAGTAA-3′) primers as detailed previously [14]. Prior to DGGE analysis, samples were analyzed via 1% agarose gel electrophoresis. 

#### 2.3.3. DGGE Analysis

The BioRad DCode Universal Mutation Detection System (BioRad, Hercules, CA, USA) was used for DGGE analyses with 8% (*w*/*v*) polyacrylamide gels in 1 × TAE. A 30–50% urea-formamide denaturing gradient (diluted from a 7 M urea and 40% (*w*/*v*) formamide stock) yielded optimal fungal sample separation. Gels were run for 17 h at 100 V at 60 °C, after which they were stained with AgNO_3_ as published previously [15]. The Quantity One software and a calibrated imaging densitometer GS-710 (Bio-Rad, Hercules, CA, USA) were then used to image and analyze DGGE fingerprint profiles. 

#### 2.3.4. DGGE Band Sequencing

Representative DGGE were excised with a sterile scalpel, and were added to ultrapure water overnight at 4 °C to facilitate sample elution. Samples from eluted bands were then again amplified with the GC-clamp primers detailed above, After amplification, samples were again assessed via DGGE gels to confirm purity. Bands were then re-amplified using the same primers without the GC clamp, and were purified using a universal PCR purification kit (Sangon, Shanghai, China). Cloning and sequencing were then performed by Sangon, and the resultant sequences were compared to ITS sequences in the GenBank (http://www.ncbi.nlm.nih.gov, accessed on 20 October 2021) databases to identify the closest phylogenetic relatives. 

### 2.4. Data Analysis

Cluster and community diversity analyses were performed with the Quantity One software, with individual DGGE lanes being converted into densitometric profiles. Fungal community Shannon–Wiener index of general diversity (H), the Evenness (E), and the richness (S) values were then calculated based on relative band intensity with the PAST software package (Palaeontology Statistics, http://folk.uio.no/ohammer/past/, accessed on 20 October 2021). The unweighted pair group method with arithmetic averages (UPGMA) was used for sample clustering. 

## 3. Results

### 3.1. DGGE Profiling of Fungal Communities

We began by characterizing the DGGE fingerprint profiles for pit mud fungal communities (Figure 2). There were clear differences in the communities present within pit mud samples from the upper wall, middle wall, lower wall, and bottom cellar layers (Table 1). The Shannon-Wiener index value for the fungal community from the middle wall layer was greater than the corresponding values for the other analyzed pit mud samples, suggesting that maximal fungal diversity was present within this middle wall layer. The evenness index (E) values for these different fungal communities were between 0.961 and 0.996, with these values being higher for samples from the middle wall and cellar bottom relative to other samples. Middle wall pit mud samples also exhibited the highest species richness index value, followed by samples from the bottom of the cellar, with no significant differences in these values when comparing samples from the upper or lower cellar wall. 

UPGMA dendrograms were constructed for DGGE profiles based on Dice coefficient values in order to describe community similarity between pit mud samples from different positions within the fermentation cellar (Figure 3). Cluster analyses of these fungal profiles revealed that pit mud samples from the upper wall layer formed a group, while the primary microbial community present in samples from the lower wall layer were similar to those in pit mud samples from the cellar bottom (Figure 3). 

To more fully understand the dominant fungi within pit mud samples, DGGE profile bands were carefully excised, purified, and sequenced (Table 2, Figure 2, Appendix A). In total, 51 bands were sequenced, with the resultant sequences having a similarity of 96% to those in the GenBank database. These ITS sequences were associated with 25 fungal genera: *Penicillium, Alternaria*, *Trichosporon*, *Simplicillium*, *Leptobacillium*, *Penicillifer*, *Calonectria*, *Ramgea*, *Aotearoamyces*, *Fusarium*, *Epicoccum*, *Bipolaris*, *Metarhizium*, *Cladosporium*, *Seltsamia*, *Malassezia*, *Aspergillus, Pichia*, *Ascochyta*, *Thermomyces*, *Antarctomyces*, *Fusarium*, *Didymella*, *Ilyonectria*, and *Candida*. The two dominant genera in these samples were *Aspergillus* and *Alternaria* species, which accounted for 21.57% and 15.69% of the identified fungi, respectively. 

Internal transcribed spacer (ITS) is a piece of nonfunctional RNA located between structural ribosomal RNAs (rRNA) of a common precursor transcript, which is especially useful for elucidating relationships among congeneric species and closely related genera [16]. ITS sequences are most informative among subgenera, and variability is low between closely related species. However, the limitation of the application technology of using ITS sequences as molecular markers is that some fungi, due to evolutionary order, variation, and even analysis methods, show little difference in the interval, which is not suitable for the markers of intrageneric species and populations. Therefore, many dominant genera in this study (e.g., *Alternaria*, *Aspergillus*, *Fusarium*, *Penicillium*) cannot be identified at the species level only with the ITS region, and it should be combined with traditional morphological methods. 

As shown in Figure 2 and Figure 4, Alternaria doliconidium (band 9), Ramgea ozimecii (band 10), Alternaria destruens (band 11), Alternaria betae-kenyensis (band 21), Cladosporium chasmanthicola (band 22), Seltsamia ulmi (band 23), and Penicillium argentinense (band 38) were present in all pit mud samples, with Alternaria destruens (band 11) and Alternaria doliconidium (band 9) being present at notably high levels, suggesting that they may be dominant members of the pit mud flora and that they may be key mediators of liquor fermentation, although additional research will be needed to test this possibility. In contrast, Penicillium fuscoglaucum (band 1), Penicillium glandicola (band 2), Aotearoamyces nothofagi (band 12), Malassezia restricta (band 28), Penicillium lanosocoeruleum (band 31), Penicillium crustosum (band 32), and Aspergillus tonophilus (band 35) were only detected in the pit mud layer form the upper cellar wall, whereas Alternaria alstroemeriae (band 3), Trichosporon insectorum (band 4), Fusarium equiseti (band 14), Calonectria pseudoreteaudii (band 20), Penicillium clavigerum (band 26), Penicillium compactum (band 30), Ascochyta phacae (band 36), Metarhizium frigidum (band 39), Alternaria burnsii (band 40), Fusarium nurragi (band 46), and Didymella keratinophila (band 47) were only present in the middle cellar wall. Similarly, Alternaria zantedeschiae (band 17), and Ilyonectria cyclaminicola (band 50) were only detected in pit mud samples from the lower cellar wall, while Leptobacillium leptobactrum (band 6), Calonectria queenslandica (band 8), Aspergillus appendiculatus (band 42), and Candida pseudolambica (band 51) were only present in samples from the bottom pit mud layer. Antarctomyces psychrotrophicus (band 45), and Aspergillus heterocaryoticus (band 48) were present at high levels in the middle wall, lower wall, and bottom pit mud layers. Trichosporon inkin (band 24) was present in all three wall layers from the same cellar, while Simplicillium chinense (band 5), Trichosporon coremiiforme (band 25), and Aspergillus intermedius (band 33) were only evident in the upper and middle wall layers. Penicillifer martinii (band 7), Fusarium circinatum (band 15), Epicoccum phragmospora (band 16), and Bipolaris axonopicola (band 18) were present in the middle wall and cellar bottom pit mud layers. Penicillium caseifulvum (band 29) was only found in the upper wall, middle wall, and cellar bottom pit mud layers, whereas Metarhizium robertsii (band 19), Penicillium roqueforti (band 27), and Pichia kudriavzevii (band 34) were present in the upper wall layer and the bottom layer. Alternaria radicina (band 41) and Alternaria radicina (band 49) were only found in the middle and lower wall layers, and Alternaria helianthiinficiens (band 13) was detected in the lower wall and bottom layers. 

### 3.2. Physiochemical Properties

The physicochemical properties of pit mud samples from different cellar positions were next evaluated (Table 3). Levels of moisture, pH, PO_4_^3−^, acetic acid, Humus, K^+^, Mg^2+^, Ca^2+^, acetic acid, butyric acid, and caproic acid, changed incrementally with position from the upper wall layer to the deepest bottom pit mud layer, consistent with the gradient-like distribution of these physicochemical attributes in 20-year-old pit mud, as previously demonstrated by Meng et al. [17]. Levels of NH^4+^-N were higher in the bottom pit mud layer relative to other layers, whereas these levels did not differ significantly between the middle and bottom wall pit mud layers, and were lowest in the upper wall layer pit mud samples. In contrast, lactic acid levels exhibited the opposite trend such that these levels were significantly lower in the bottom pit mud wall layer. 

### 3.3. Relationships between Fungal Communities and Physicochemical Variables

A redundancy analysis (RDA) was next conducted to better clarify potential relationships between the 51 detected fungal genera and the 12 analyzed environmental factors (Figure 5). The first two component axes explained 77.6% of the variation in fungal composition, with species-specific environmental correlations for both axes of 48.1% and 78.6%, respectively, indicating that fungal community structure was moderately correlated with these physicochemical variables. An interactive forward selection procedure was used to evaluate these 12 environmental variables, revealing that moisture, pH, and NH_4_^+^-N contributed significantly to community composition (39.5%, 13.8%, and 13.8%, respectively; *p* < 0.01), whereas the other eight variables exhibited no significant correlations. 

## 4. Discussion

Chinese strong-flavour liquor is prepared through the fermentation of a mixture of sorghum, rice, and wheat known as *Zaopei* in a rectangular cellar composed of pit mud. This pit mud is an ideal habitat for microbes that are integral to the distillation process, serving as key determinants of the flavour of the resultant liquor. The quality of pit mud is thus an important regulator of the quality and taste of the liquor produced. 

Pit mud tends to age with increasing cellar usage, and the microbial communities present within this mud vary based upon their spatial location within the walls or bottom of the cellar. A range of sensory descriptions and physicochemical indices have been used to describe pit mud from different locations within these fermentation cellars. For example, pit mud from the bottom of these cellars is often described as smooth, fine, soft, moist, and sooty with an aroma of esters, ammonia, and hydrogen sulfide. In contrast, pit mud from the top of these cellars is rough, hard, dry, and light grey with white lumps or aciform crystals and no aroma. While pit mud from the bottom layer can support the production of good-quality liquor, that from the upper layer cannot. As such, studying the microbial communities present within pit mud is essential in order to understand the molecular mechanisms governing the flavour and aroma of Chinese strong-flavour liquor in an effort to improve the quality of this popular and culturally important beverage. 

In prior studies, researchers have utilized both culture-dependent and –independent strategies to determine that bacteria, fungal, archaea, and actinomycetes species are present within pit mud samples, with bacteria and archaea being dominant in this environment [18]. At the family level, common pit mud-resident bacteria include *haloplasmatacea*, *Bacillaceae*, *planococcaceae*, *synergistaceae*, *staphylococcaceae*, *Thermoanaerobacter*, and *clostridiaceae* species. Archaea present within pit mud are largely consistent across regions, and primarily include methanobacteria (*Methanobacteriaceae*), *Methanococcus* (*Methanococcus*), and *thermoplasmataceae* (*thermoplasmata*) species [18]. Microbes in the *Clostridia* class are thought to be primary producers of short- and medium-chain fatty acids including butanoic acid and hexanoic acid, which are directly relevant to the liquor production process [4]. Liu et al. isolated the *Lysi-nibacillussphaerieus*, *Brevibacillusbrevis*, and *Paenibacilluslarvae subsup. pulvifaciens* strains from pit mud and found that these microbes were important mediators of fermentation and producers of aromatic compounds in the context of Chinese strong-flavour liquor distillation [19]. Wang et al. explored bacterial community structures in samples of pit mud from a 16-year-old Chinese strong-flavour liquor cellar, and found that *Clostridium*, *Aminobacterium*, *Petrimonas*, *Syntrophmonas*, and *Sedimentibacter* species were the dominant microbes therein [20]. Ding et al. employed a PCR-DGGE approach to characterize the eubacterial pit mud communities associated with Luzhou-flavour liquor and consistently detected higher levels of eubacterial diversity in samples from the bottom of the cellar relative to samples from the cellar walls [8]. Most of these past studies have specifically focused on prokaryotic species, and there have been few comparable analyses of the fungal communities found within pit mud. 

Herein, we explored the structures of fungal communities in multidimensional pit mud environments via a DGGE approach, revealing clear discrimination between the communities present in different locations within the fermentation cellar. *Penicillium roqueforti*, *Pichia kudriavzevii*, *Aotearoamyces nothofagi*, *Penicillium robsamsonii*, *Alternaria arborescens*, *Trichosporon insectorum*, *Seltsamia ulmi*, *Trichosporon coremiiforme*, *Malassezia restricta* were dominant in the pit mud samples form the upper cellar wall, whereas *Metarhizium frigidum*, *Calonectria pseudoreteaudii*, *Penicillium clavigerum*, *Fusarium equiseti*, *Simplicillium chinense*, *Aspergillus intermedius*, *Trichosporon coremiiforme*, *Fusarium circinatum*, *Alternaria radicina*, *Aspergillus heterocaryoticus* were dominant in the middle cellar wall. *Alternaria radicina*, *Cladosporium chasmanthicola*, *Alternaria helianthiinficiens*, *Penicillium argentinense*, *Antarctomyces psychrotrophicus*, *Trichosporon inkin* majorly fungus presented in the down cellar wall layer. *Bipolaris axonopicola*, *Ramgea ozimecii*, *Penicillium argentinense*, *Calonectria queenslandica*, *Metarhizium robertsii*, and *Penicillium roqueforti* were identified as the dominated fungal in pit mud samples from the cellar bottom. Additionally, *Alternaria destruens* and *Alternaria doliconidium* are present at notably high levels in all layers of pit mud samples. These differences may explain why the quality of strong-flavour liquor varies with cellar position. We found that fungal abundance in the upper and middle layers was significantly higher than that in the lower wall and bottom layers, potentially due to the lower oxygen levels in these latter two environments, as such oxygen deficiency may have compromised fungal survival [9]. This, in turn, may explain the higher saccharification efficiency that is typically detected in the upper and middle *Zaopei* layers in the context of liquor fermentation. 

It was reported that *Penicillium roqueforti* has a high lypolytic activity, which may play an important function in esterification during the fermentation of Chinese strong flavour liquor [21]. *Penicillium argentinense*, *Penicillium robsamsonii*, and *Penicillium roqueforti* are all tannase-producing strains [22]. *Aspergillus intermedius*, and *Aspergillus heterocaryoticus* are saccharifying enzyme-producing strains [23]. *Pichia kudriavzevii* is a potential producer of bioethanol and phytase, which was commonly presented in the cellar of Chinese strong flavour liquor [24]. *Trichosporon coremiiforme* is reported to be a microbial oil-producing strain [25], which might help to improve the flavour of Chinese strong flavour liquor. However, the functions of *Aotearoamyces nothofagi*, *Alternaria arborescens*, *Alternaria radicina*, *Alternaria destruens*, *Alternaria helianthiinficiens*, *Malassezia restricta*, *Metarhizium frigidum*, *Calonectria pseudoreteaudii*, *Calonectria queenslandica*, *Fusarium equiseti*, *Fusarium circinatum*, *Cladosporium chasmanthicola*, *Antarctomyces psychrotrophicus*, *Bipolaris axonopicola*, *Ramgea ozimecii*, and *Metarhizium robertsii* on the brewing of Chinese strong flavour liquor are still unclear, and needs further research. 

With respect to pit mud physicochemical properties, we found that moisture, pH, PO4^3−^, acetic acid, Humus, K^+^, Mg^2+^, Ca^2+^, acetic acid, butyric acid, and caproic acid levels rose with sample position from the upper wall to the bottom of the fermentation cellar, suggesting that organic compounds were gradually degraded with the position. The maximal moisture levels in the bottom pit mud layer may be associated with the high levels of *Huangshui* present in this setting. The higher pH levels lower in the cellar may be linked to the degradation of various acids such as lactic acid [20], and the synthesis of ammonium nitrogen, consistent with the observed trends in NH_4_^+^-N levels. The lower acetic acid levels with the upper wall pit mud layer are consistent with less robust prokaryotic metabolism in this location, given that acetic acid is a metabolic end product produced by many bacterial species [4]. The rising lactic acid levels detected from the bottom of the pit to the upper pit may correspond to the different *Lactobacillus* activity levels in these positions. 

We then conducted an RDA analysis to explore relationships between the 51 detected fungal genera and the 12 measured environmental variables (Figure 4). This revealed that moisture, pH, and NH_4_^+^-N levels were the most significant environmental factors, accounting for 67.1% of microbial community variability, indicating that these physicochemical factors are closely linked to pit mud microbe growth. Fungal community structure was moderately correlated with these physicochemical variables. For example, *Alternaria zantedeschiae*, *Ilyonectria cyclaminicola*, *Calonectria pseudoreteaudii*, *Leptobacillium leptobactrum*, *Calonectria queenslandica*, *Aspergillus appendiculatus*, *Aspergillus heterocaryoticus*, *Penicillium argentinense*, *Antarctomyces psychrotrophicus*, and *Ramgea ozimecii* levels were strongly positively correlated with moisture, pH, NH_4_^+^-N, PO_4_^3−^, Humus, K^+^, Mg^2+^, Ca^2+^, acetic acid, butyric acid, and caproic acid levels, whereas for *Metarhizium robertsii*, *Alternaria destruens*, *Bipolaris axonopicola*, *Thermomyces lanuginosus*, and *Cladosporium chasmanthicola* levels these correlations were more moderate. 

Many prior studies have sought to understand the relationship between pit mud physicochemical properties and the microbial communities therein. Meng et al., for example, found that these properties were significantly influenced by depth within the fermentation cellar [17]. Zhang et al. found that acid and amino nitrogen concentrations were higher in the bottom pit mud layers relative to other positions, suggesting that these compounds may influence the overall diversity of the microbial communities found within this bottom layer [4]. We similarly detected a clear relationship between fungal community structure and physicochemical variables in pit mud samples. However, further research will be essential to develop the efficient cultivation strategies necessary to delineate the independent contributions of different fungi to the production of Chinese strong-flavour liquor production.

## 5. Conclusions 

This study explored the multidimensional distributions of fungal communities and physicochemical properties in different spatial positions of pit mud by using PCR-DGGE methods. There were clear differences in the fungal communities present within pit mud samples from the upper wall, middle wall, lower wall, and bottom cellar layers. RDA analysis demonstrated that a clear relationship between fungal community structure and physicochemical variables in different spatial pit mud samples, especially moisture, pH, and NH_4_^+^-N were identified as the three most significant factors associated with the fungal community through a redundancy analysis. This study provides theoretical basis to design effective strategies to manipulate microbial consortia for better improving pit mud quality in Chinese strong-flavour liquor production.

## Figures and Tables

**Figure 1 foods-11-03544-f001:**
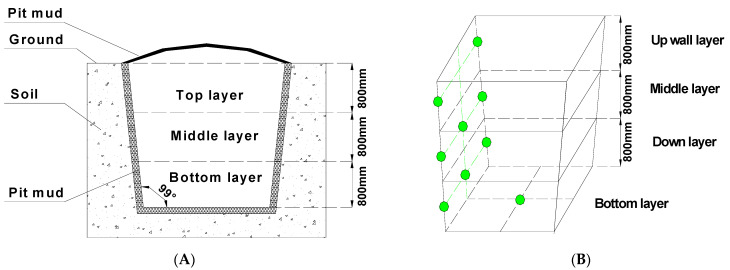
The profiles of the Chinese strong-flavour liquor pit mud (**A**) and the sampling sites of pit mud (**B**). The green dot means each pit mud sampling site of cellar.

**Figure 2 foods-11-03544-f002:**
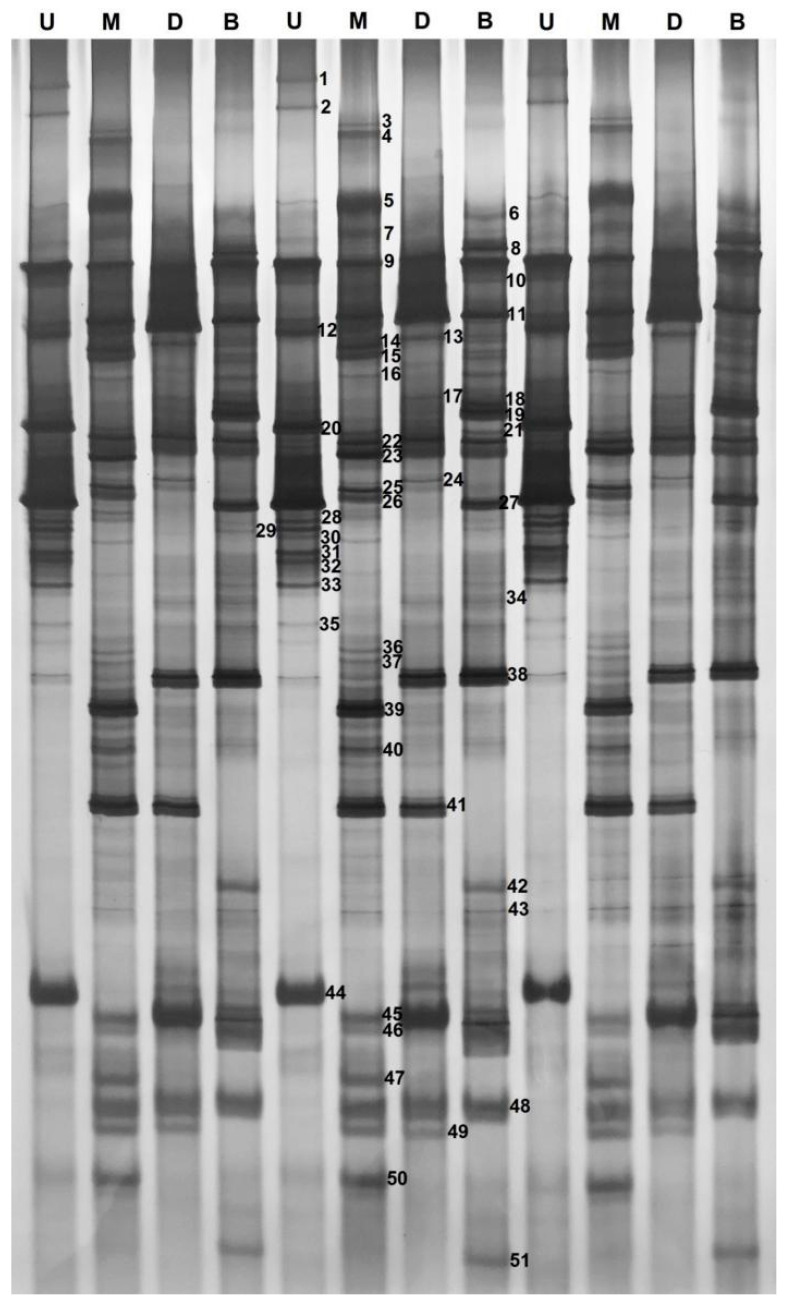
PCR-DGGE fingerprints of ITS gene extracted from the fungal community in the pit mud samples collected from different spatial positions of the cellar. Lanes U, M, D, and B represent samples collected from the up wall layer of the cellar, middle wall layer of the cellar, down wall layer of the cellar, and bottom layer of the cellar, respectively. The bands indicated with numbers were excised and sequenced and the alignment results are listed in Table 2.

**Figure 3 foods-11-03544-f003:**
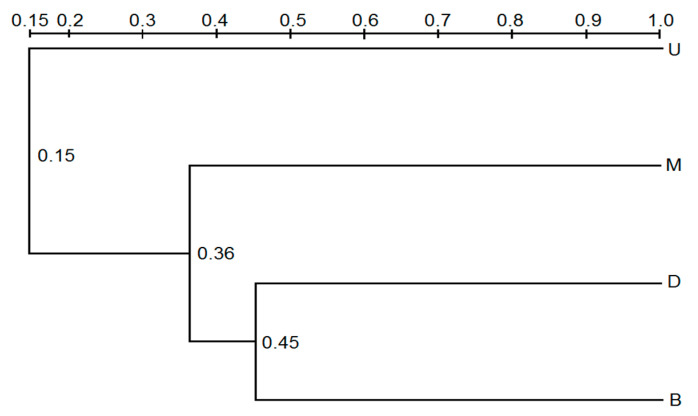
Clustering analysis of fungal DGGE profiles. The similarity was calculated using Euclidean distance and clustering was performed using UPGMA. U, M, D, and B represent samples from the up wall layer of cellar, middle wall layer of cellar, down wall layer of cellar, and bottom layer of cellar.

**Figure 4 foods-11-03544-f004:**
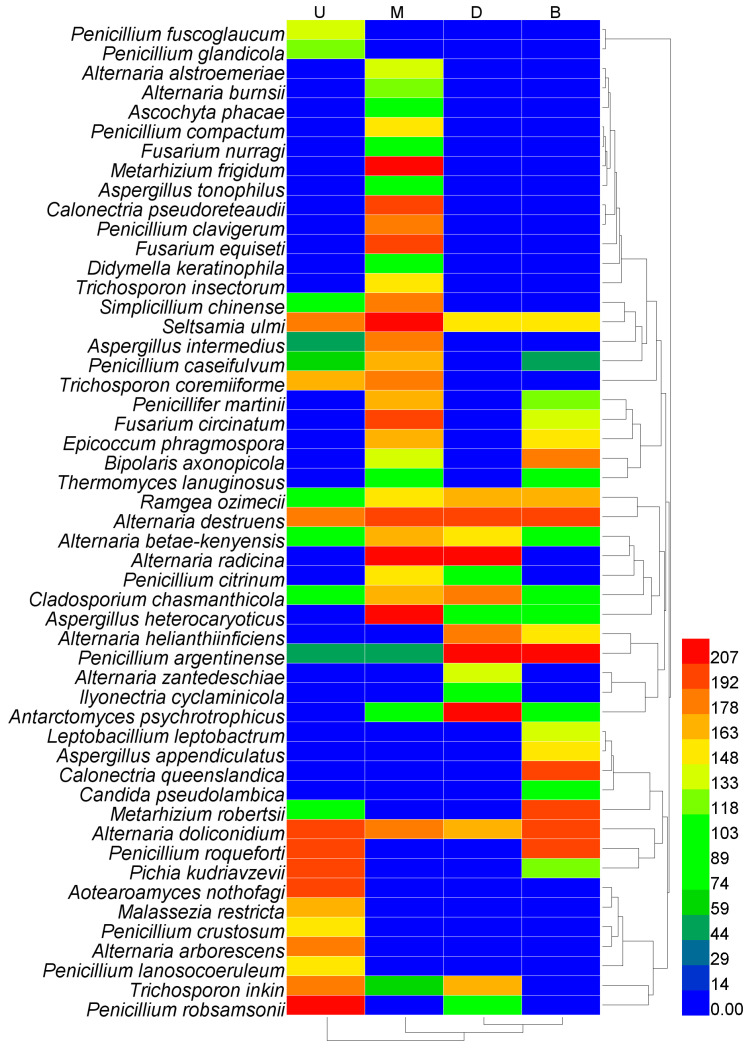
Heatmap of fungal communities in the pit mud samples from different spatial positions of the cellar. Lanes U, M, D, and B respectively represent pit mud samples collected from up wall layer of cellar, middle wall layer of cellar, down wall layer of cellar, and bottom layer of cellar, and were sampled from the same fermentation cellar. The scale bar shows the abundance of the genera, red and blue represents high and low abundance, respectively. The tree diagram shows the cluster analysis results of different fungal communities from different spatial positions in the cellar.

**Figure 5 foods-11-03544-f005:**
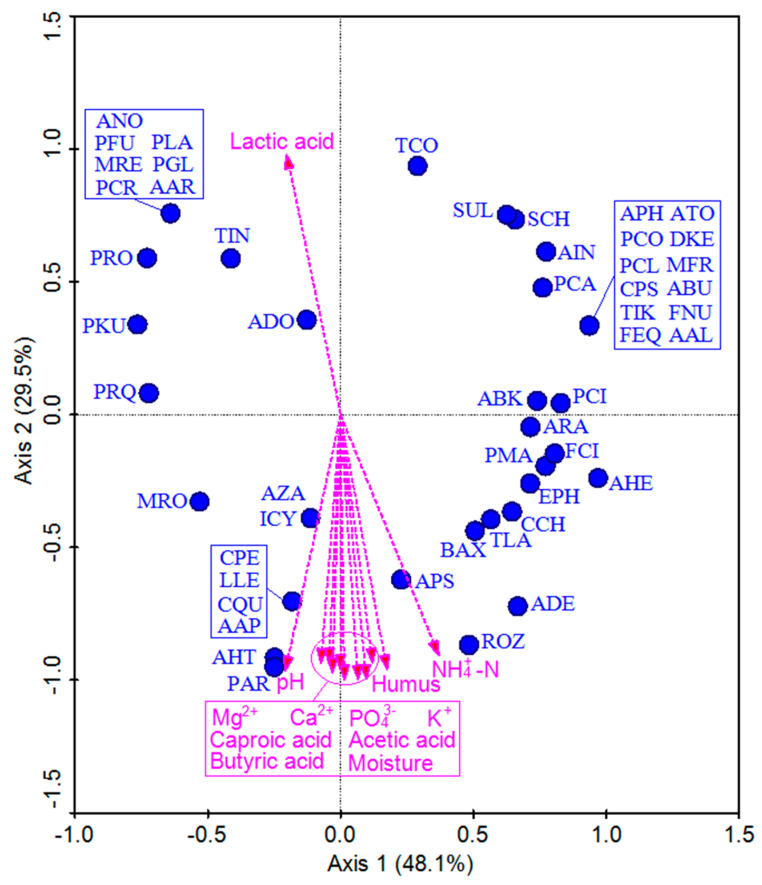
Redundancy analysis of fungal communities and physico-chemical attributes. The arrows indicate the direction and magnitude of biogeochemical attributes associated with fungal community structures. AAL: *Alternaria alstroemeriae*, AAP: *Aspergillus appendiculatus*, AAR: *Alternaria arborescens*, ABK: *Alternaria betae-kenyensis*, ABU: *Alternaria burnsii*, ADE: *Alternaria destruens*, ADO: *Alternaria doliconidium*, AHE: *Alternaria helianthiinficiens*, AHT: *Aspergillus heterocaryoticus*, APH: *Ascochyta phacae*, APS: *Antarctomyces psychrotrophicus*, AIN: *Aspergillus intermedius*, ANO: *Aotearoamyces nothofagi*, ARA: *Alternaria radicina*, ATO: *Aspergillus tonophilus*, AZA: *Alternaria zantedeschiae*, BAX: *Bipolaris axonopicola*, CCH: *Cladosporium chasmanthicola*, CPS: *Calonectria pseudoreteaudii*, CPE: *Candida pseudolambica*, CQU: *Calonectria queenslandica*, DKE: *Didymella keratinophila*, EPH: *Epicoccum phragmospora*, FEQ: *Fusarium equiseti*, FCI: *Fusarium circinatum*, FNU: *Fusarium nurragi*, ICY: *Ilyonectria cyclaminicola*, LLE: *Leptobacillium leptobactrum*, MFR: *Metarhizium frigidum*, MRE: *Malassezia restricta*, MRO: *Metarhizium roberts*, PAR: *Penicillium argentinense*, PCA: *Penicillium caseifulvum*, PCI: *Penicillium citrinum*, PFU: *Penicillium fuscoglaucum*, PCL: *Penicillium clavigerum*, PCO: *Penicillium compactum*, PCR: *Penicillium crustosum*, PRO: *Penicillium robsamsonii*, PRQ: *Penicillium roqueforti*, PGL: *Penicillium glandicola*, PKU: *Pichia kudriavzevii*, PLA: *Penicillium lanosocoeruleum*, PMA: *Penicillifer martinii*, ROZ: *Ramgea ozimecii*, SCH: *Simplicillium chinense*, SUL: *Seltsamia ulmi*, TCO: *Trichosporon coremiiforme*, TIN: *Trichosporon insectorum*, TIK: *Trichosporon inkin*, TLA: *Thermomyces lanuginosus*. As shown in Figure 5, AZA (*Alternaria zantedeschiae*), ICY (*Ilyonectria cyclaminicola*), CPE (*Calonectria pseudoreteaudii*), LLE (*Leptobacillium leptobactrum*), CQU (*Calonectria queenslandica*), AAP (*Aspergillus appendiculatus*), AHT (*Aspergillus heterocaryoticus*), PAR (*Penicillium argentinense*), APS (*Antarctomyces psychrotrophicus*), and ROZ (*Ramgea ozimecii*) were strongly positively correlated with moisture, pH, NH_4_^+^-N, PO_4_^3−^, Humus, K^+^, Mg^2+^, Ca^2+^, acetic acid, butyric acid, and caproic acid. In addition, MRO (*Metarhizium robertsii*), ADE (*Alternaria destruens*), BAX (*Bipolaris axonopicola*), TLA (*Thermomyces lanuginosus*), and CCH (*Cladosporium chasmanthicola*) were moderately positively correlated with these variables, while correlations were weaker for EPH (*Epicoccum phragmospora*), PMA (*Penicillifer martinii*), AHE (*Alternaria helianthiinficiens*), and FCI (*Fusarium circinatum*). As shown in the upper portion of Figure 5, TIN (*Trichosporon inkin*), and ADO (*Alternaria doliconidium*) were closely associated with lactic acid, while ANO (*Aotearoamyces nothofagi*), PFU (*Penicillium fuscoglaucum*), PLA (*Penicillium lanosocoeruleum*), MRE (*Malassezia restricta*), PGL (*Penicillium glandicola*), PCR (*Penicillium crustosum*), AAR (*Alternaria arborescens*), PRO (*Penicillium robsamsonii*), PRQ (*Penicillium roqueforti*), PKU (*Pichia kudriavzevii*), TCO (*Trichosporon coremiiforme*), SUL (*Seltsamia ulmi*), SCH (*Simplicillium chinense*), AIN (*Aspergillus intermedius*), PCA (*Penicillium caseifulvum*), APH (*Ascochyta phacae*), ATO (*Aspergillus tonophilus*), PCO (*Penicillium compactum*), DKE (*Didymella keratinophila*), PCI (*Penicillium citrinum*), MFR (*Metarhizium frigidum*), CPS (*Candida pseudolambica*), ABU (*Alternaria burnsii*), TIN (*Trichosporon insectorum*), FNU (*Fusarium nurragi*), FEQ (*Fusarium equiseti*), and AAL (*Alternaria alstroemeriae*) were only weakly correlated with this variable.

**Table 1 foods-11-03544-t001:** Indices of fungal diversity in the samples collected from different spatial positions of the cellar according to quantified bands from Figure 2.

Lane ^a^	Shannon-Wiener	Evenness	Richness
U	3.17	0.989	25
M	3.69	0.996	41
D	3.15	0.993	24
B	3.45	0.996	32

^a^ Lanes U, M, D, and B respectively represent pit mud samples collected from the up wall layer of the cellar, middle wall layer of the cellar, down wall layer of the cellar, and bottom layer of the cellar, and were sampled from the same fermentation cellar.

**Table 2 foods-11-03544-t002:** BLAST Identified gene sequences of ITS—derived bands excised from a DGGE gel.

Band No. ^a^	Closest Relative (NCBI Accession No.)	Identity (%) ^b^
1	*Penicillium fuscoglaucum* (NR_163669.1)	97.25
2	*Penicillium glandicola* (MH860946.1)	97.40
3	*Alternaria alstroemeriae* (MH863036.1)	99.61
4	*Trichosporon insectorum* (MW433667.1)	98.54
5	*Simplicillium chinense* (MK102638.1)	100.00
6	*Leptobacillium leptobactrum* (MG786580.1)	97.04
7	*Penicillifer martinii* (KJ869167.1)	96.19
8	*Calonectria queenslandica* (NR_121455.1)	97.59
9	*Alternaria doliconidium* (MT672468.1)	100.00
10	*Ramgea ozimecii* (KY368752.1)	96.94
11	*Alternaria destruens* (DQ323680.1)	100.00
12	*Aotearoamyces nothofagi* (MG807392.1)	96.79
13	*Alternaria helianthiinficiens* (MF414166.1)	96.42
14	*Fusarium equiseti* (KX463025.1)	99.59
15	*Fusarium circinatum* (NR_120263.1)	96.14
16	*Epicoccum phragmospora* (MW237699.1)	96.92
17	*Alternaria zantedeschiae* (MH864493.1)	96.66
18	*Bipolaris axonopicola* (KX452443.1)	97.56
19	*Metarhizium robertsii* (NR_132011.1)	96.21
20	*Calonectria pseudoreteaudii* (NR_137040.1)	96.64
21	*Alternaria betae-kenyensis* (NR_136118.1)	98.19
22	*Cladosporium chasmanthicola* (NR_152307.1)	100.00
23	*Seltsamia ulmi* (NR_156634.1)	96.38
24	*Trichosporon inkin* (NR_073243.1)	98.51
25	*Trichosporon coremiiforme* (NR_073249.1)	98.03
26	*Penicillium clavigerum* (NR_121317.1)	96.52
27	*Penicillium roqueforti* (NR_103621.1)	100.00
28	*Malassezia restricta* (NR_103585.1)	98.88
29	*Penicillium caseifulvum* (NR_163685.1)	96.34
30	*Penicillium compactum* (NR_144844.1)	96.33
31	*Penicillium lanosocoeruleum* (NR_163541.1)	96.78
32	*Penicillium crustosum* (NR_077153.1)	96.69
33	*Aspergillus intermedius* (NR_137448.1)	99.01
34	*Pichia kudriavzevii* (NR_131315.1)	98.15
35	*Alternaria arborescens* (NR_135927.1)	100.00
36	*Ascochyta phacae* (KT389475.1)	96.62
37	*Aspergillus tonophilus* (NR_137450.1)	97.54
38	*Penicillium argentinense* (NR_121523.1)	96.89
39	*Metarhizium frigidum* (NR_132012.1)	96.02
40	*Alternaria burnsii* (NR_136119.1)	99.10
41	*Alternaria radicina ATCC* (NR_165503.1)	97.18
42	*Aspergillus appendiculatus* (NR_135433.1)	96.91
43	*Thermomyces lanuginosus* (NR_121309.1)	99.69
44	*Penicillium robsamsonii* (NR_144866.1)	96.58
45	*Antarctomyces psychrotrophicus* (NR_164292.1)	97.47
46	*Fusarium nurragi* (NR_159860.1)	97.75
47	*Didymella keratinophila* (NR_158275.1)	97.72
48	*Aspergillus heterocaryoticus* (NR_163674.1)	100.00
49	*Penicillium citrinum* (NR_121224.1)	99.50
50	*Ilyonectria cyclaminicola* (NR_121495.1)	97.27
51	*Candida pseudolambica* (NR_153281.1)	97.53

^a^ Numbers are those of bands shown in Figure 2. ^b^ Most homologous BLAST-derived match.

**Table 3 foods-11-03544-t003:** The physicochemical properties of pit mud samples from different spatial positions of the cellar.

Parameter	U	M	D	B
Moisture (%)	32.54 ± 2.65 ^a^	35.11 ± 1.51 ^b^	37.68 ± 2.57 ^c^	39.35 ± 2.15 ^d^
pH	5.23 ± 0.25 ^a^	5.45 ± 0.16 ^a^	7.56 ± 0.46 ^b^	9.23 ± 0.56 ^c^
NH^4+^-N (g/kg)	2.06 ± 0.21 ^a^	3.98 ± 0.29 ^b^	4.01 ± 0.35 ^b^	5.28 ± 0.37 ^c^
PO_4_^3−^ (mg/kg)	201.35 ± 15.32 ^a^	256.35 ± 20.31 ^b^	335.26 ± 28.35 ^c^	387.65 ± 30.21 ^d^
Humus (%)	5.35 ± 0.34 ^a^	9.024 ± 0.87 ^b^	10.31 ± 0.89 ^c^	15.56 ± 1.32 ^d^
K^+^ (mg/kg)	525.35 ± 46.72 ^a^	678.54 ± 52.08 ^b^	834.21 ± 54.32 ^c^	1125.35 ± 67.25 ^d^
Mg^2+^ (mg/kg)	134.65 ± 69.17 ^a^	181.45 ± 56.23 ^b^	201.32 ± 68.45 ^c^	245.32 ± 78.65 ^d^
Ca^2+^ (mg/kg)	368.32 ± 13.54 ^a^	438.57 ± 25.21 ^b^	517.36 ± 23.56 ^c^	708.19 ± 47.43 ^d^
Acetic acid (mg/kg)	556.54 ± 46.28 ^a^	677.35 ± 58.32 ^b^	856.37 ± 75.64 ^c^	1235.94 ± 98.56 ^d^
Butyric acid (mg/kg)	397.86 ± 32.82 ^a^	623.74 ± 58.08 ^b^	926.48 ± 86.37 ^c^	1021.87 ± 90.89 ^d^
Caproic acid (mg/kg)	2356.54 ± 120.37 ^a^	3570.35 ± 234.52 ^b^	5256.37 ± 136.85 ^c^	7563.25 ± 163.21 ^d^
Lactic acid (mg/kg)	25,348.89 ± 875.89 ^d^	18,692.32 ± 785.65 ^c^	13,897.87 ± 567.31 ^b^	11,783.41 ± 710.65 ^a^

Note: (1) all samples mean air-dry samples. (2) U, M, D, and B respectively represent pit mud samples collected from up wall layer of cellar, middle wall layer of cellar, down wall layer of cellar, and bottom layer of cellar, and were sampled from the same fermentation cellar. (3) All data are presented as means ± standard deviations, different small letters in the same column represent significant differences at 0.05 level.

## Data Availability

The data presented in this study are available in the article and Appendix A.

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
