# Peer review of "Fungal Diversity Profiles in Pit Mud Samples from Chinese Strong-Flavour Liquor Pit"

_foods, 2022, doi:10.3390/foods11223544_

Round 1
Reviewer 1 Report
Dear authors,
I have enjoyed the manuscript titled “Fungal community diversity profiles in pit mud samples from 2 Chinese Strong-flavour liquor pit”. This manuscript provided valuable data on the fungal community found in the fermentation cell of Chinese liquor and their relationship with chemical properties in the fermentation cell. Please find the minor points below that would improve the quality of the manuscript.
Fig. 4: Please add a description of the tree on the right side of the heatmap. Is this a phylogenetic tree of fungal taxa identified from the pit mud sample, or was there another analysis? In addition, please explain the meaning of the figure legend (band intensity?).
Table 2: Please re-adjust the spacing within the table.
L515-518: Please check the paragraph spacing between references.
Author Response
Dear authors,
I have enjoyed the manuscript titled “Fungal community diversity profiles in pit mud samples from 2 Chinese Strong-flavour liquor pit”. This manuscript provided valuable data on the fungal community found in the fermentation cell of Chinese liquor and their relationship with chemical properties in the fermentation cell. Please find the minor points below that would improve the quality of the manuscript.
Fig. 4: Please add a description of the tree on the right side of the heatmap. Is this a phylogenetic tree of fungal taxa identified from the pit mud sample, or was there another analysis? In addition, please explain the meaning of the figure legend (band intensity?).
Now, we have already added a description of the tree on the right side of the heatmap. The tree diagram in Figure 4 shows the cluster analysis results of different fungal communities from different spatial positions of cellar. In addition, the meaning of the figure legend has also been explained in Figure 4. The scale bar shows the abundance of the genera, red and blue represents high and low abundance, respectively.
Table 2: Please re-adjust the spacing within the table.
Now, we have already re-adjusted the spacing within the table.
L515-518: Please check the paragraph spacing between references.
Now, we have already re-adjusted the paragraph spacing between references according to the requirements of the manuscript.
Reviewer 2 Report
This study investigated the fungal diversity in pit mud samples from the fermentation cellars producing Chinese strong-flavor liquor. The fungal community structure was analyzed by the PCR-DGGE method and showed a relationship with physicochemical properties. The dominant fungal species were significantly different between layers and some properties showed a significant relationship with these species. The subject is interesting but the manuscript needs to be revised well.
1. In the Abstract, there is none of the backgrounds but a lot of fungal species are listed. This is not a part of the Results, thus it needs to be summarized well. Please add a background of the study and remove the list of the fungal species.
2. Sampling strategy is not clearly explained. There is a figure and description, but there is none of the numbers of cellar and samples used in the final analysis. For example, there are 9 or 17 (9+7) dots in Figure 1, but the number of lanes is 12 (4 layers x 3 replicates?). How the samples were combined or not?
3. The structure of Results and Discussion is more like “Result” and “Discussion”, not “Results and Discussion”. It should be revised well; These can be separated in this form, or keeps it by merging discussions to the end of the description of each result.
4. Many dominant genera in this study (e.g. Alternaria, Aspergillus, Fusarium, Penicillium) cannot be identified at the species level only with the ITS region. It needs to be discussed with the limit of study.
5. There is none of the statistical tests for the physicochemical properties. Please add it to Table 3.
Title: Fungal “community diversity profiles” is too redundant. Remove “community” or “diversity”. Please use the lower letter for the “S”trong.
Line 21: fungal -> fungi.
L38-39: Change “spp.” to non-italics.
L68, 72: Italics for Clostridium.
L88: Fungal communities not “eukaryotic communities”.
L91: Remove “.” Between mud and by.
L92-93: Remove “fungal community” between multidimensional and distribution.
L132: Use the full name of ITS at the first time.
L132-135: The original reference of the primers should be added.
L187: What is NFTSW?
L196: Community, not population.
L226-257: I think a Venn diagram is needed in this part.
Figure 1: Please revise the figure to show the word clearly. What is different in the color between light green and blue dots (Fig 1B)?
Figure 4: What unit of measurement is used for the color? What is the meaning of the numbers?
Author Response
This study investigated the fungal diversity in pit mud samples from the fermentation cellars producing Chinese strong-flavor liquor. The fungal community structure was analyzed by the PCR-DGGE method and showed a relationship with physicochemical properties. The dominant fungal species were significantly different between layers and some properties showed a significant relationship with these species. The subject is interesting but the manuscript needs to be revised well.
- In the Abstract, there is none of the backgrounds but a lot of fungal species are listed. This is not a part of the Results, thus it needs to be summarized well. Please add a background of the study and remove the list of the fungalspecies.
Now, we have already added the backgrounds in the Abstract part.
- Sampling strategy is not clearly explained. There is a figure and description, but there is none of the numbers of cellar and samples used in the final analysis. For example, there are 9 or 17 (9+7) dots in Figure 1, but the number of lanes is 12 (4 layers x 3 replicates?). How the samples were combined or not?
Now, we have already added the Sampling strategy “Each sample plot was divided into four different positions (upper layer of the cellar wall, middle layer of the cellar wall, down layer of the cellar wall and the bottom of the cellar), and the pit mud wall samples were collected from the center of each wall, with approximately 100 g of mud being collected per position and mixed to yield a composite sample. Samples of pit mud from the cellar bottom were collected from the center of each pit. All samples were collected at a depth of ~5 cm. All samples were well mixed, then transferred to sterile polyethylene bags and stored at -20 °C prior to analysis.” in the Methods parts, and Figure 1 has also been modified in the manuscript.
- The structure of Results and Discussion is more like “Result” and“Discussion”, not “Results and Discussion”. It should be revised well; These can be separated in this form, or keeps it by merging discussions to the end of the description of each result.
Now, we have already separated the Discussion from the parts of “Results and Discussion” in the manuscript.
- Many dominant genera in this study (e.g. Alternaria, Aspergillus, Fusarium, Penicillium) cannot be identified at the species level only with the ITS region. It needs to be discussed with the limit of study.
We have already discussed the limit of ITS sequences in the manuscript.
- There is none of the statistical tests for the physicochemical properties. Please add it to Table 3.
Now, we have already added the statistical tests for the physicochemical properties in Table 3.
6.Title: Fungal “community diversity profiles” is too redundant. Remove “community” or “diversity”. Please use the lower letter for the “S”trong.
Now, we have already removed “community” in the title, and used the lower letter for the “S”trong.
- Line 21: fungal -> fungi.
Now, we have replaced “fungi” by the word of “fungal” in Line 21.
- L38-39: Change “spp.” to non-italics.
Now, we have already changed “spp.” to non-italics in L38-39.
- L68, 72: Italics for Clostridium.
Now, we have already changed “Clostridium” to Italics styles in L68, 72.
- L88: Fungal communities not “eukaryotic communities”.
Now, we have already changed “eukaryotic communities” to “Fungal communities” in L88.
- L91: Remove “.” Between mud and by.
Now, we have already removed “.” Between mud and by in L91.
- L92-93: Remove “fungal community” between multidimensional and distribution.
Now, we have already removed “fungal community” between multidimensional and distribution in L92-93.
- L132: Use the full name of ITS at the first time.
Now, we have already used the full name of ITS at the first time in L132.
- L132-135: The original reference of the primers should be added.
The primers have already been referenced in L132-135.
- L187: What is NFTSW?
Now, we have already deleted the mistaken sentence of “Lane N represent samples collected from NFTSW” in L187.
- L196: Community, not population.
Now, we have already changed “population” to “Community” in L196.
- L226-257: I think a Venn diagram is needed in this part.
In this paragraph, we have already used RDA analysis to explore relationships between the detected fungal genera and the measured environmental variables, therefore, we think it is not necessary to use a Venn diagram in this part.
- Figure 1: Please revise the figure to show the word clearly. What isdifferent in the color between light green and blue dots (Fig 1B)?
Now, we have already revised the figure in Figure 1, and the word has already showed clearly.
21.Figure 4: What unit of measurement is used for the color? What is the meaning of the numbers?
The color of red and blue represents high and low abundance of the genera, respectively in figure 4, which has already been explained in the figure legend.